# Factors Related to Depression Associated with Chewing Problems in the Korean Elderly Population

**DOI:** 10.3390/ijerph18116158

**Published:** 2021-06-07

**Authors:** Hyejin Chun, Miae Doo

**Affiliations:** 1Department of Family Medicine, CHA Bundang Medical Center, CHA University, Gyeonggi 13496, Korea; hyejinchun@chamc.co.kr; 2Department of Food and Nutrition, Kunsan National University, Jeonbuk 54150, Korea

**Keywords:** chewing problem, depression, dietary protein consumption, elderly, Korean National Health and Nutrition Examination Survey

## Abstract

Unlike younger adults, depression in older adults is sometimes related to chewing problems. This study examined the risk factors related to depression associated with chewing problems in 3747 elderly individuals using the Korean National Health and Nutrition Examination Survey. Approximately 41.2% of the total subjects reported chewing problems. There were significant differences in age, education, marital status, individual income, current smoking status, and aerobic physical activity in relation to chewing problems (*p* < 0.001 for all). The subjects who experienced chewing problems showed a higher score on the EuroQoL 5 Dimension index (*p* < 0.001) but a lower health-related quality of life than those with no chewing problems (*p* < 0.001). The prevalence of depression, which was classified by the patient health questionnaire—9, in subjects with chewing problems was approximately 2 times higher than that in those with no chewing problems (*p* < 0.001). Subjects with chewing problems were found to have a 1.945-fold higher adjusted risk of depression than those who did not have chewing problems (95% CI = 1.583–2.390, *p* < 0.001), and subjects with high protein consumption showed a 1.410-fold greater risk of depression (95% CI = 1.144–1.739, *p* = 0.001) than those with low protein consumption.

## 1. Introduction

The elderly population is increasing worldwide. According to “World Population Prospects 2019,” published by the United Nations, the number of persons aged 65 years or over reached 703 million in 2019 and is predicted to increase to 1.5 billion in 2050 [1]. In other words, it is predicted that the population will change from 1 in 11 persons are aged 65 or older in 2019 to 1 in 6 persons in 2050 [1]. The elderly population in South Korea is 8.03 million, which means that, currently, 1 in 6 persons is elderly according to data from Statistics Korea [2]. As the elderly population increases, maintaining the health of the elderly is recognized as a public problem.

Elderly individuals experience various physiological changes, including cardiovascular, renal, pulmonary, gastrointestinal, endocrine, immune, and neurological functions, with aging [3]. Aging could be affected by the presence or absence of chronic diseases, drug use, chewing conditions, and psychological status [4,5]. In particular, chewing problems are a more common symptom in the elderly population and are a result of oral problems such as tooth loss, caries, and ill-fitting dentures [6]. Poor oral health in older individuals is associated with poor nutritional status [5,7,8], increasing the severity of chronic diseases and mortality [5,8,9,10]. Chewing difficulty is also related to adverse psychological aspects, such as subjective well-being or quality of life [6,11,12]. In the same context, an association between chewing problems and depression in the elderly has been reported [8,13].

Depression in older adults, recognized as a serious public health problem, is different from that in younger adults [14,15]. Many studies have reported that depression in older adults is more likely to be affected by the presence of cognitive disorders and chronic diseases, such as dementia, stroke, Parkinson’s disease, diabetes, and cardiovascular disease [16,17]. In addition, socioeconomic variables such as retirement, social isolation, poor economic status, and bereavement are closely related to depression in the elderly [15,18]. Additionally, depression in older adults is less common than that in younger adults, but serious adverse outcomes occur, including increased chronic disease, cognitive impairment, reduced functioning, and an increased risk of suicide [15].

Although some have reported a relationship between chewing problems and depression [8,9,10,11,12,13] and others have reported the risk factors for depression in older adults [15,16,17,18], the risk of depression due to chewing problems and the various risk factors affecting these associations have not been clearly documented in a large-scale elderly population. Therefore, in this study, the patient health questionnaire—9 (PHQ-9) was used as a quick and easy tool for the detection of depression and the measurement of its severity [19]. Additionally, this study examined the prevalence and risk of depression according to chewing problems, and these associations were found to be influenced by various factors among the Korean elderly individuals who participated in the Korean National Health and Nutrition Examination Survey (KNHANES).

## 2. Subjects and Methods

### 2.1. Study Design and Subjects Selection

This study used data from the KNHANES VI (VI-1, 2014) and VII (VII-1, 2016, and VII-3, 2018) surveys because the PHQ-9 was measured every two years [20]. The KNHANES is a nationally representative survey designed to investigate health and nutritional status; it has been conducted by the Korean Centers for Disease Control and Prevention (KCDC) [21]. The KNHANES consists of health interviews, health examinations, and nutritional surveys. Among noninstitutionalized Korean citizens from South Korea, the subjects in the KNHANES participated based on a stratified multistage clustered probability design. Detailed information on the survey design of the KNHANES has been reported elsewhere [21]. Among 23,692 subjects (7550 in 2014, 8150 in 2016, and 7992 in 2018) in the KNHANES, 3747 subjects (weighted *n* = 6,348,653 including 2,883,195 men and 3,465,458 women) who were aged 65 years and older with no missing data concerning their chewing status- and depression-related variables and reported a plausible daily total energy consumption of ≥500 kcal and ≤5000 kcal were selected for this study. The KNHANES was approved by the Institutional Review Board of KCDC, and all participants provided written informed consent. Before analyzing the data from KNHANES, our study was approved by the Institutional Review Board of Kunsan National University (IRB No. 1040117-202008-HR-015-01).

### 2.2. Measurements

Chewing status, lifestyle-related variables, health-related quality of life, depression degree, and dietary macronutrient consumption were collected from the KNHANE data. In the health interviews in the KNHANE, chewing status, lifestyle-related variables, health-related quality of life, and depression degree were surveyed by trained specialists using questionnaires. Dietary macronutrient consumption data from the nutritional survey were obtained from face-to-face interviews conducted by trained dietitians.

### 2.3. Chewing Status

Current chewing status was assessed based on the following question: “Are you currently feeling uncomfortable chewing food due to problems in your mouth such as teeth, dentures, or gums?” The subjects were divided into “chewing problem (CP),” including “very uncomfortable” or “uncomfortable,” groups; and “no chewing problem (NCP),” including “not bad,” “not very uncomfortable,” or “not at all uncomfortable,” groups according to their answer [7].

### 2.4. Sociodemographic Variables

As sociodemographic variables, age, educational level, marriage status, individual income, current alcohol consumption, current smoking status, and aerobic physical activity were collected using questionnaires. The educational levels were divided into “≤ middle school” or “≥ high school.” Marital status was divided into “married” as the current marriage status and “unmarried,” including never married, divorced, and widowed. Individual income was classified into “low” and “high” based on their income’s median. The subjects were classified as a “regular drinker” or “nondrinker” based on their alcohol consumption during the past 1 year and a “current smoker” or “nonsmoker” based on their current smoking pattern. Usual aerobic physical activity was divided into “yes” or “no” based on activity during the previous week.

### 2.5. Depression

The subjects’ depression was assessed using the PHQ-9. The PHQ-9 is a self-reported screening tool to measure the presence and severity of depression [19]. It comprises 9 depressive questions about the last 2 weeks, with a 4-point Likert scale. The overall scores ranged from 0 to 27, with higher scores indicating more depression. Depression was defined, in accordance with the cutoff point of the Korean version of the PHQ-9, as a score of 5 or greater [22].

### 2.6. Health-Related Quality of Life

Health-related quality of life (HRQoL) was assessed using the Korean version of the EuroQoL 5 Dimension (EQ-5D) [23]. The EQ-5D consists of five health dimensions: mobility, self-care, usual activities, pain/uncomfortable, and anxiety/depression. The three response levels of the problem were answered for each dimension as follows: “no problem,” “moderate problem,” and “extreme problem.” Therefore, 243 (3^5^) response levels of EQ-5D indicate different health statuses. These were converted into the EQ-5D index as a single numeric score using a weight model for the Korean population [24]. The EQ-5D index scores ranged from −0.171 to 1, with lower scores indicating poor health status [25]. The cutoff point of the EQ-5D index score for the severity assessment of major depression is 0.673 based on a previous study in a Korean population [26].

### 2.7. Assessment of Dietary Macronutrients Consumption

Nutrition surveys were carried out by trained dietitians in the form of face-to-face interviews [21]. Dietary macronutrient consumption was assessed using a 24 h recall record survey. Dietary macronutrient consumption, including carbohydrate, protein, and fat consumption, was calculated based on the food composition database published by the Rural Development Administration of Korea [27].

### 2.8. Statistical Analyses

The data were analyzed using SPSS (version 24.0; IBM Corp., Armonk, NY, USA) software for Windows. All statistical analyses applied the sampling weights to reflect estimates of the entire Korean population. The data are presented as estimated percentages (standard errors (SEs)) for categorical variables or estimated means (95% confidence intervals (95% CIs)) for continuous variables. Pearson’s chi-square test was used to identify the sociodemographic variables, depression, and EQ-5 dimensions (mobility, self-care, usual activities, pain/discomfort, anxiety/depression) according to chewing problems. Continuous variables, such as age, PHQ-9, EQ-5D index, and dietary macronutrient consumption, were analyzed by performing independent *t*-tests. To identify the risk of depression depending on the chewing problem without adjusting for the covariates, logistic regression analysis was used. Multinomial logistic regression models after adjusting for the covariates were analyzed to estimate the risk of depression based on chewing problems. As covariates, sociodemographic variables (gender, age, educational level, individual income, and marriage status), health-related variables (regular drinker, current smoker, and aerobic physical activity), health-related quality of life, and dietary protein consumption were used. Statistical significance was defined as a *p*-value < 0.05.

## 3. Results

The sociodemographic characteristics of the participants stratified by chewing problems are presented in Table 1. The average age was 72.65 years (72.44–72.87), and 41.2% of the subjects experienced chewing problems (SE = 1.0%). Subjects who experienced chewing problems were significantly older (*p* < 0.001). Education level, marital status, individual income, current smoking status, and aerobic physical activity were significantly different between the CP and NCP groups (*p* < 0.001 for all). Regular drinking had marginal differences for chewing problems (*p* = 0.049). However, there was no difference in chewing problems by gender (*p* = 0.070).

The HRQoL measured using the EQ-5D dimension and dietary macronutrient consumption are shown by chewing problems in Table 2 and Table 3. All EQ-5D dimensions according to chewing problems were significantly different. Among the EQ-5D dimension, 47.2% of the subjects with chewing problems also suffered mobility problems, whereas 29.2% of the subjects with no chewing problems suffered mobility problems (*p* < 0.001). Significant differences in other dimensions in relation to chewing problems were found (*p* < 0.001 for all). The proportions of the CP group having any problems were 14.9% with self-care, 28.2% with usual activities, 46.4% with pain/discomfort, and 21.5% with anxiety/depression. Those of the NCP group were 5.9%, 13.7%, 27.4%, and 10.4%, respectively. The subjects with chewing problems had higher EQ-5D index scores (0.92 (0.91–0.92) vs. 0.84 (0.83–0.85)) than those with no chewing problems. The percentages of the CP and NCP groups with poor health-related quality of life, which was defined with the EQ-5D index using 0.673 as the cutoff point, were 11.6% and 3.3%, respectively (*p* < 0.001).

Chewing problems were associated with dietary macronutrient consumption. In other words, the subjects with chewing problems consumed less energy (1617.63 kcal vs. 1723.95 kcal, *p* < 0.001), carbohydrates (284.31 g vs. 297.29 g, *p* = 0.004), protein (52.40 g vs. 57.18 g, *p* < 0.001), and fat (24.85 g vs. 27.76 g, *p* < 0.001) than those with no chewing problems.

The depression-related variables according to chewing problems are presented in Figure 1. Significant differences were observed between the CP and NCP groups. The subjects who experienced chewing problems had significantly higher total PHQ-9 scores for measuring depression than those with no chewing problems ((3.59 (3.30–3.88) vs. 2.05 (1.87–2.23), *p* < 0.001, Figure 1A)). The prevalence of depression in the CP group was approximately 2 times higher than that in the NCP group ((28.5% vs. 14.4%, *p* < 0.001), Figure 1B)). Additionally, the risk for depression in relation to chewing problems was 2.370-fold (95% CI = 1.983–2.832, *p* < 0.001) higher in the CP group than in the NCP group (Figure 1C).

A multinomial logistic regression analysis was performed to identify the factors affecting depression caused by chewing problems after adjusting for covariates (Table 4). As a result, significant differences in the adjusted risk for depression were observed in relation to chewing problems, gender, education level, marriage status, individual income, health-related quality of life, and dietary protein consumption. The subjects experiencing chewing problems had a 1.945-fold higher adjusted risk of depression than those who did not experience chewing problems (95% CI = 1.583–2.390, *p* < 0.001). In terms of gender, women had a 2.206-times greater risk of depression than men (95% CI = 1.672–2.911, *p* < 0.001). The risk of depression was 1.332 times greater in subjects with low income than subjects with high income (95% CI = 1.075–1.650, *p* = 0.009). The subjects reporting low HRQoL had a 5.405-fold higher adjusted risk of depression than those who had a high HRQoL (95% CI = 3.968–7.353, *p* < 0.001). The subjects who consumed low levels of dietary protein had a 1.410-fold greater risk of depression (95% CI = 1.144–1.739, *p* = 0.001).

## 4. Discussion

This study used data from the KNHANES, a large-scale nationally representative survey of the Korean elderly population, to examine the factors related to depression and chewing problems. The sociodemographic variables, HRQoL measured using the EQ-5D dimension, and dietary macronutrient consumption significantly differed between CP and NCP. Additionally, the prevalence and risk of depression were associated with chewing status. Interestingly, we identified that being women, having a low income, poor HRQoL, low consumption of dietary protein, and chewing problems were all significantly associated with a higher risk of depression.

A total of 41.2% of the subjects suffered chewing problems in our study. This prevalence of chewing problems is higher than in Chinese (37.8%) [6], Japanese (39.0%) [8], and Italian populations (31.6%) [13]. This study revealed that chewing problems are related to sociodemographic variables, such as age, education level, marriage status, individual income, and health-related variables, such as current smoking status and aerobic physical activity. Subjects with chewing problems were older, which is consistent with previous studies that reported that satisfaction with chewing ability decreases with aging [18,28]. Additionally, elderly subjects with chewing problems had a lower educational level and economic conditions, were current smokers, and were less likely to engage in aerobic physical activity than those with no chewing problems, which agrees with the results of other studies [6,13].

Many previous studies have found that chewing ability is correlated with age-related diseases in older populations [5,8,9,10,28]. Although the association between chewing difficulty and other diseases was not examined, HRQoL was associated with chewing status in our study. HRQoL, which was measured using the EQ-5D, reflects an individual’s health status from the perspective of quality of life and has been recognized to influence the presence and severity of disease in terms of its impacts on physical impairment and functional status [29]. Therefore, the HRQoL could be used to understand the current health status due to all diseases in older populations. The results of our study showed that elderly individuals who experienced chewing problems had a higher prevalence of poor health-related quality of life (11.6% for the CP group vs. 3.3% for the NCP group) than those who did not experience chewing problems. Interestingly, among the EQ-5D dimension, mobility and pain/discomfort were experienced at high proportions of 47.2% and 46.4% in subjects with chewing problems, which is approximately 2 times the proportion of those with no chewing problems. Our findings indicate that elderly individuals with chewing problems are more likely to have or develop age-related diseases. Many previous studies have reported that chewing problems are related to food selection and diversity and this in turn influences nutritional quality [5,6,7]. These results were similar to our findings and showed that subjects with chewing problems had lower consumption of dietary macronutrients. In the same context, generally, older populations with chewing problems are prone to a lack of some nutrients, such as proteins, vitamins, or minerals, which increases their risk of malnutrition, decreased physical function, and age-related disease [5,30]. Our results suggest that chewing problems could deteriorate their nutritional status and decrease their health-related quality of life and that their subsequent nutritional status and health-related quality of life could simultaneously worsen their chewing ability.

The subjects with chewing problems showed a higher depression prevalence and an increased risk for depression than those with no chewing problems. These findings are similar to previous studies in which elderly individuals who experience chewing problems are associated with an increased risk of depression [6,8]. However, the assessment tools for depression used in previous studies were different from the one used in our study. In other words, in our study, depression in the elderly, which differs from that in younger adults, was defined using the PHQ-9. A meta-analysis by Levis et al. reported that the PHQ-9 is sensitive for screening to detect major depression and is more specific for older subjects than for younger subjects [31]. In addition, our study identified the factors affecting depression caused by chewing problems. The increased risk for depression according to chewing problems was associated with HRQoL, dietary protein consumption, and sociodemographic variables, such as gender, education level, marriage status, and individual income. As mentioned earlier, poor nutritional status and HRQoL due to chewing problems could act as co-risk factors for depression. In particular, consumption of dietary protein is related to HRQoL. Many studies have argued that sufficient and adequate consumption of dietary protein is related to increased muscle mass [32,33], and individuals with sarcopenia, defined by low muscle mass and function, have a lower HRQoL and depression [34,35]. Moreover, a systematic review by Tamura et al. reported that low body weight (with low body muscle mass), which is caused by poor oral condition, including chewing problem, eating dependency, and dysphagia, is related to depression [36]. These results suggest that chewing problems could be a factor that exacerbates depression through deteriorating dietary conditions and a poor subjective quality of life.

This study demonstrated an association between chewing problems and depression, which was assessed by the PHQ-9 in a large-scale elderly population. However, our findings have several limitations. First, the data from the KNHANES used in this study are cross-sectional, so the results of this study should not be used to definitely support causal relationships. Second, because the self-reported variable of chewing problems was used, objective chewing ability should be identified considering various oral health-related variables, such as the number of teeth, cavities, and the use of dentures. Third, the presence or severity of depression among the elderly population is affected by socioeconomic and health-related variables, especially the presence of several age-related diseases. In this study, several factors related to depression due to chewing problems were identified among various socioeconomic and health-related factors. Especially, HRQoL, one of the risk factors, is related to age-related diseases in the elderly population [37]. However, chewing problem or depression could be directly affected or complicated by cognitive disorders and chronic diseases; therefore, future studies need to consider age-related diseases as risk factors.

## 5. Conclusions

Using large-scale data of the elderly population from the KNHANES, our study demonstrated that chewing problems are associated with a risk of depression. These associations are affected by HRQoL and dietary protein consumption as well as sociodemographic factors. The findings from our study suggest that improving HRQoL or proper consumption of dietary protein could potentially modify the risk of depression associated with chewing problems.

## Figures and Tables

**Figure 1 ijerph-18-06158-f001:**
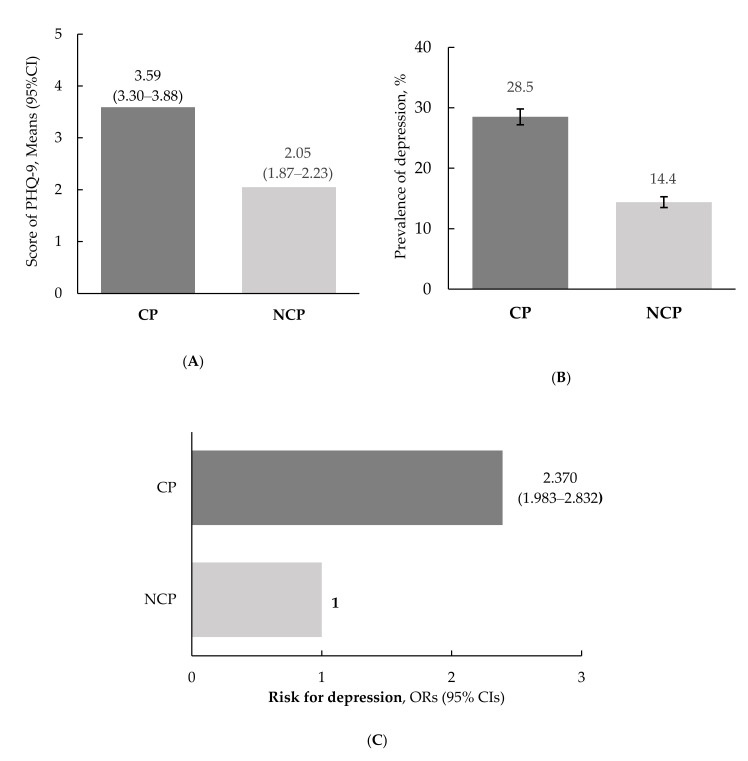
Depression according to chewing problems. (**A**) Scores on the patient health questionnaire-9. The data are presented as estimated means (95% CIs), and the *p*-value for comparisons of variables in relation to chewing problems was assessed using a *t*-test (*p* < 0.001). (**B**) Prevalence of depression. The data are presented as estimated percentages (SE), and the *p*-value between chewing problems was assessed using the x^2^-test (*p* < 0.001). (**C**) Risk of depression. The data are presented as ORs (95% CIs), and the *p*-value between chewing problems was assessed using logistic regression analysis. CP, chewing problem; NCP, no chewing problem; PHQ-9, patient health questionnaire-9; ORs, odds ratios; 95% CIs, confidence intervals; SE, standard error.

**Table 1 ijerph-18-06158-t001:** Sociodemographic characteristics by chewing problems in Korean elderly individuals.

Sociodemographic Characteristics	CP (*n* = 1596)	NCP (*n* = 2151)	*p*-Value *
Gender, men	43.4 (1.4)	46.8 (1.1)	0.070
Age, year	73.21 (27.92–73.51)	72.10 (71.81–72.38)	<0.001
Education level, ≥high school	21.3 (1.5)	34.6 (1.5)	<0.001
Marriage status, married	63.2 (1.5))	71.1 (1.2)	<0.001
Individual income, high	49.6 (1.7)	54.9 (1.5)	<0.001
Regular drinker	34.2 (1.4)	37.9 (1.2)	0.049
Current smoker	11.9 (1.0)	8.0 (0.8)	0.001
Aerobic physical activity, yes	30.5 (1.4)	39.5 (1.4)	<0.001

The data are presented as estimated percentages (standard errors) for categorical variables or estimated means (95% confidence intervals) for continuous variables. CP, chewing problem; NCP, no chewing problem. * The *p*-values between chewing problems were assessed using the x^2^-test or *t*-test.

**Table 2 ijerph-18-06158-t002:** Health-related quality of life by chewing problems in elderly Koreans.

Health-Related Quality of Life	CP (*n* = 1596)			NCP (*n* = 2151)			*p*-Value *
	Response Levels of the Problem						
EQ-5D Dimension	None	Moderate	Extreme	None	Moderate	Extreme	
Mobility	52.8 (1.5)	44.2 (1.5)	3.0 (0.5)	70.8 (1.2)	28.8 (1.2)	0.4 (0.1)	<0.001
Self-care	85.1 (1.1)	13.4 (1.0)	1.5 (0.4)	94.1 (0.6)	5.5 (0.6)	0.4 (0.1)	<0.001
Usual activities	71.8 (1.4)	26.2 (1.4)	2.0 (0.4)	86.3 (0.9)	13.3 (0.9)	0.4 (0.1)	<0.001
Pain/discomfort	53.6 (1.4)	37.2 (1.4)	9.2 (0.9)	72.6 (1.2)	24.4 (1.2)	3.0 (0.4)	<0.001
Anxiety/depression	78.5 (1.1)	18.2 (1.1)	3.3 (0.5)	89.6 (0.8)	9.6 (0.8)	0.8 (0.2)	<0.001
EQ-5D index	0.92 (0.91–0.92)			0.84 (0.83–0.85)			<0.001
Poor HRQoL **	11.6 (1.0)			3.3 (0.4)			<0.001

The data are presented as estimated percentages (standard errors) for categorical variables or estimated means (95% confidence intervals) for continuous variables. CP, chewing problem; NCP, no chewing problem.; EQ-5D, EuroQoL 5 Dimension; HRQoL, health-related quality of life. * The *p*-values between chewing problems were assessed using the x^2^-test or *t*-test. ** Poor HRQoL was defined with the EQ-5D index using 0.673 as the cutoff point.

**Table 3 ijerph-18-06158-t003:** Dietary macronutrient consumption by chewing problems in elderly Koreans.

Dietary Consumption	CP (*n* = 1596)	NCP (*n* = 2151)	*p*-Value *
Energy (Kcal)	1617.63 (1580.07–1655.19)	1723.95 (1690.42–1757.48)	<0.001
Carbohydrate (g)	284.31 (277.71–290.91)	297.29 (291.19–303.38)	0.004
Protein (g)	52.40 (50.82–53.98)	57.18 (55.80–58.56)	<0.001
Fat (g)	24.82 (23.57–26.07)	27.76 (26.74–28.77)	<0.001

The data are presented as estimated means (95% confidence intervals). CP, chewing problem; NCP, no chewing problem. * The *p*-values between chewing problems were assessed using a *t*-test.

**Table 4 ijerph-18-06158-t004:** Factors related to depression by chewing problems in elderly Koreans.

Factors Related to Depression	Reference	ORs (95% CI)	*p*-Value *
Chewing problem	No	1.945 (1.583–2.390)	<0.001
Gender	Men	2.206 (1.672–2.911)	<0.001
Age	65~74 years	0.856 (0.685–1.070)	0.172
Education level	≥High school	1.001 (0.764–1.312)	0.992
Marriage status	Married	1.240 (0.982–1.567)	0.070
Individual income	High	1.332 (1.075–1.650)	0.009
Regular drinker	No	0.910 (0.727–1.137)	0.405
Current smoker	No	1.365 (0.950–1.961)	0.092
Aerobic physical activity	Yes	0.892 (0.703–1.132)	0.346
HRQoL	High	5.405 (3.968–7.353)	<0.001
Dietary protein consumption	High	1.410 (1.144–1.739)	0.001

The data are presented as ORs (95% CIs). * The ORs (95% CIs) and the *p*-values were calculated in reference to no chewing problems using multinomial logistic regression analysis after adjusting for covariates. Covariates included gender, age, education level, marriage status, individual income, regular drinker, current smoker, aerobic physical activity, health-related quality of life and dietary protein consumption. ORs, odds ratio; 95% Cis, confidence intervals.

## Data Availability

The data used in this study is available from the Korea Center for Disease Control and Prevention, following webpage: https://knhanes.cdc.go.kr/ (accessed on 7 June 2021).

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
