# Peer review of "Factors Related to Depression Associated with Chewing Problems in the Korean Elderly Population"

_ijerph, 2021, doi:10.3390/ijerph18116158_

Round 1

Reviewer 1 Report

The authors point to chewing disturbance being associated with depression.  This is an important avenue to pursue further research regarding potential preventative measures to reduce the risk of depression such as attention to sufficient dietary protein intake.  The methodology is sound as there is a large sample selected from a national study with clear inclusion criteria.

Author Response

We appreciated the reviewer for careful reading and description about our manuscript with the valuable comments.

Reviewer 2 Report

Very intriguing manuscript.
Chewing is one of the problems of aging that mainly promotes nutritional disorders . There is no doubt that the poor social and economic condition does not allow to dedicate oneself to the care of dental problems. This aggravates the masticatory function and favors the onset of depression.
The authors have described very well the problems and correctly analyzed the results.
One of the solutions could be the intervention of the public health assistance in the free treatment of dental pathology.

Author Response

(The authors gave the same response as above.)

Reviewer 3 Report

  1. The study addresses the issue of depression associated with chewing problems. There is a systematic review on this subject that the authors should cite. [Tamura BK, Bell CL, Masaki KH, Amella EJ. Factors associated with weight loss, low BMI, and malnutrition among nursing home patients: a systematic review of the literature. J Am Med Dir Assoc. 2013;14(9):649-55. doi: 10.1016/j.jamda.2013.02.022] .
  2. Economic status was not one of the socio-demographic factors that the authors considered. This data could be important because financial resources are essential to obtain dental care and solve chewing problems.
  3. Depression was assessed with a self-completed questionnaire and not with a specialist assessment. The diagnosis is therefore not certain. Authors should more correctly speak of "risk of depression" rather than "major depression".
  4. Poverty is a powerful risk factor for depression. The study did not examine this and, as a result, the association between chewing problems and depression could be a spurious association. The authors should discuss this limitation.

Author Response

We appreciated the reviewer for careful reading and description about our manuscript with the valuable comments. We worked to the best of my abilities to revise the issues reviewer point out.

  1. The study addresses the issue of depression associated with chewing problems. There is a systematic review on this subject that the authors should cite. [Tamura BK, Bell CL, Masaki KH, Amella EJ. Factors associated with weight loss, low BMI, and malnutrition among nursing home patients: a systematic review of the literature. J Am Med Dir Assoc. 2013;14(9):649-55. doi: 10.1016/j.jamda.2013.02.022] .

As you suggested, we have been cited and reviewed in DISCUSSION section, as follow; line 290-292.

“Also, a systematic review by Tamura et al. reported that low body weight (with low body muscle mass), which is caused by poor oral condition including chewing problem, eating dependency, dysphagia, is related with depression [36].”

  1. Economic status was not one of the socio-demographic factors that the authors considered. This data could be important because financial resources are essential to obtain dental care and solve chewing problems.

We appreciate the constructive and very helpful comments. Like your comment, we agreed that the economic conditions was one of important factors related to depressive disorder or chewing problem. Therefore, we reanalyzed and rewritten about economic conditions, as follow; SUBJECTS AND METHODS sections, line 104-105, and 147-150.

“Individual income was classified into “low” or “high” based on their income’s median.”

“. As covariates, sociodemographic variables (gender, age, educational level, individual income, and marriage status), health-related variables (regular drinker, current smoker, and aerobic physical activity), health-related quality of life, and dietary protein consumption were used”

RESULTS sections, line 155-157, and 209-221, Table 1 and 4.

“Education level, marital status, individual income, current smoking status, and aerobic physical activity were significantly different between the CP and NCP groups (p < 0.001 for all).”

“A multinomial logistic regression analysis was performed to identify the factors affecting depression caused by chewing problems after adjusting for covariates (Table 4). As a result, significant differences in the adjusted risk for depression were observed in relation to chewing problems, gender, education level, marriage status, individual income, health-related quality of life, and dietary protein consumption. The subjects experiencing chewing problems had a 1.945-fold higher adjusted risk of depression than those who did not experience chewing problems (95% CI= 1.583-2.390, p < 0.001). In terms of gender, women had a 2.206-times greater risk for depression than men (95% CI= 1.672-2.911, p < 0.001). The risk of depression was 1.332 times greater in subjects with low income than subjects with high income (95% CI= 1.075-1.650, p = 0.009). The subjects reporting low HRQoL had a 5.405-fold higher adjusted risk for depression than those who a high HRQoL (95% CI = 3.968-7.353, p < 0.001). The subjects who consumed low levels of dietary protein had a 1.410-fold greater risk of depression (95% CI= 1.144-1.739, p = 0.001).”

DISCUSSION sections, line 245-248

“Additionally, elderly subjects with chewing problems had a lower educational level and economic conditions, were current smokers and were less likely to engage in aerobic physical activity than those with no chewing problems, which agrees with the results of other studies [6, 13].

  1. Depression was assessed with a self-completed questionnaire and not with a specialist assessment. The diagnosis is therefore not certain. Authors should more correctly speak of "risk of depression" rather than "major depression".

Thank you for comments.

All of manuscript have been modified according to new definition; changed from major depressive disorder to depression, as follow; line 114-116, 196-200, 209-221, and Table 4

“Depression was defined in accordance with the cutoff point of the Korean version of the PHQ-9 as a score of 5 or greater [22].”

“Han, C.; Jo, S.A.; Kwak, J.H.; Pae, C.U.; Steffens, D.; Jo, I.; Park, M.H. Validation of the Patient Health Questionnaire-9 Korean version in the elderly population: the Ansan Geriatric study. Compr Psychiatry. 2008, 49, 218-223. “

“ The prevalence of depression in the CP group was approximately 2 times higher than that in the NCP group [(28.5% vs. 14.4%, p < 0.001), Figure 1 (b)]. Additionally, the risk for depression in relation to chewing problems was 2.370-fold (95% CI = 1.983-2.832, p < 0.001) higher in the CP group than in the NCP group [Figure 1 (c)].

 “ A multinomial logistic regression analysis was performed to identify the factors affecting depression caused by chewing problems after adjusting for covariates (Table 4). As a result, significant differences in the adjusted risk for depression were observed in relation to chewing problems, gender, education level, marriage status, individual income, health-related quality of life, and dietary protein consumption. The subjects experiencing chewing problems had a 1.945-fold higher adjusted risk of depression than those who did not experience chewing problems (95% CI= 1.583-2.390, p < 0.001). In terms of gender, women had a 2.206-times greater risk for depression than men (95% CI= 1.672-2.911, p < 0.001). The risk of depression was 1.332 times greater in subjects with low income than subjects with high income (95% CI= 1.075-1.650, p = 0.009). The subjects reporting low HRQoL had a 5.405-fold higher adjusted risk for depression than those who a high HRQoL (95% CI = 3.968-7.353, p < 0.001). The subjects who consumed low levels of dietary protein had a 1.410-fold greater risk of depression (95% CI= 1.144-1.739, p = 0.001).”

  1. Poverty is a powerful risk factor for depression. The study did not examine this and, as a result, the association between chewing problems and depression could be a spurious association. The authors should discuss this limitation.

Thank you for your comment.

We has been reanalyzed and rewritten considering individual income as an indicator of poverty

Round 2

Reviewer 3 Report

The manuscript has been improved and can be published